# Transcriptome Analysis Reveals the Immunoregulation of Replacing Fishmeal with Cottonseed Protein Concentrates on *Litopenaeus vannamei*

**DOI:** 10.3390/ani13071185

**Published:** 2023-03-28

**Authors:** Hongming Wang, Xin Hu, Jian Chen, Hang Yuan, Naijie Hu, Beiping Tan, Xiaohui Dong, Shuang Zhang

**Affiliations:** 1College of Fisheries, Guangdong Ocean University, Zhanjiang 524000, China; wanghongming97@163.com (H.W.);; 2Key Laboratory of Aquatic, Livestock and Poultry Feed Science and Technology in South China, Ministry of Agriculture, Zhanjiang 524000, China; 3Aquatic Animals Precision Nutrition and High Efficiency Feed Engineering Research Center of Guangdong Province, Guangdong Ocean University, Zhanjiang 524000, China; 4Guangdong Provincial Key Laboratory of Aquatic Animal Disease Control and Healthy Culture, Guangdong Ocean University, Zhanjiang 524088, China

**Keywords:** cottonseed protein concentrate, immunoregulation, transcriptome, *Litopenaeus vannamei*

## Abstract

**Simple Summary:**

Previous study showed that moderate amounts of CPC in place of fishmeal can improve the growth performance of *Litopenaeus vannamei*, but there have been limited investigations of the mechanism of the immunological response to CPC substitution for fishmeal. In this study, high-throughput sequencing analysis was employed to investigate the changes at the gene transcription level, aiming to clarify the mechanism by which CPC substitution for fishmeal affects the immunological response in *L. vannamei*. The results showed that moderate amounts of CPC in the diet significantly improved the non-specific immune activity and expression of *L. vannamei*, which may be due to the fact that CPC increases the expression of AMP genes by inhibiting the expression of cactus genes, which may ultimately improve the immunity of *L. vannamei*. The results of this study contribute to understanding the mechanism of CPC as an alternative to fishmeal and provide useful information for the development of novel protein sources in shrimp feed.

**Abstract:**

Cottonseed protein concentrate (CPC) is a new non-food protein source with high crude protein, low price, and abundant resources, making it an ideal substitute for fishmeal. In this study, we investigated the effects of CPC re placing fishmeal on the immune response of *Litopenaeus vannamei* using transcriptome sequencing. *L. vannamei* (initial body weight: 0.42 ± 0.01 g) were fed four isonitrogenous and isolipid feeds for eight weeks, with CPC replacing fishmeal at 0% (control, FM), 15% (CPC15), 30% (CPC30), and 45% (CPC45), respectively. At the end of the feeding trial, the changes of the activities and expression of immune-related enzymes were consistent in *L. vannamei* in the CPC-containing group when compared with the FM group. Among them, the activities of ACP, PO, and LZM in the group whose diet was CPC30 were significantly higher than those in the FM group. Moreover, the activities of AKP, SOD, and CAT were significantly higher in the group containing CPC than in the FM group. Furthermore, all CPC groups had considerably lower MDA levels than the FM group. This suggests that the substitution of fishmeal with CPC leads to a significant immune response in *L. vannamei*. Compared with the FM group, transcriptome analysis identified 805 differentially expressed genes (DEGs) (484 down and 321 up), 694 (266 down and 383 up), and 902 (434 down and 468 up) in CPC15, CPC30, and CPC45, respectively. Among all DEGs, 121 DEGs were shared among different CPC-containing groups compared with the FM group. Most of these differential genes are involved in immune-related signaling pathways. The top 20 signaling pathways enriched for differential genes contained toxoplasmosis, pathogenic *Escherichia coli* infection, insulin resistance, and Toll and immune deficiency (IMD) pathways, in which NF-kappa-B inhibitor Cactus were involved. In addition, trend analysis comparison of the DEGs shared by the group with CPC in the diet and the FM group showed that Cactus genes were significantly down-regulated in the group with CPC in the diet and were lowest in the CPC30 group. Consistently, the expression of antimicrobial peptide genes was significantly higher in both diet-containing CPC groups than in the FM group. In conclusion, the moderate amount of CPC substituted for fishmeal may improve the immunity of *L. vannamei* by suppressing the expression of Cactus genes, thereby increasing the expression of antimicrobial peptide (AMP) genes.

## 1. Introduction

*Litopenaeus vannamei*, commonly known as the South American white shrimp, belongs to the Arthropoda, Crustacea, Decapoda, Penaeid, and *Penaeus* [1]. It is native to the eastern Pacific Ocean in northern Peru and Sonora, Mexico, and was introduced to China in 1988. Shrimp is currently one of China’s most important breeding species because of its delicious meat and high yield per unit [2]. With intensive farming of *L. vannamei*, compound feed gradually became the main production cost of aquaculture [3,4]. Fishmeal is considered the greatest protein source in aquatic animals because of its rich and balanced nutritional components and good palatability as a high-quality protein source for compound feeds [5]. Due to the impact of overfishing and environmental degradation, fishmeal production has declined yearly [6]. Many researchers have been committed to identifying alternative high-quality protein sources for fishmeal, and the promotion and use of sustainable high-quality protein sources have become an inevitable trend in the development of the aquaculture industry [7,8,9,10,11,12].

Cottonseed protein concentrate (CPC) is made from high quality cottonseed, with the production process of high temperature steaming and frying replaced by the technology of shell–kernel separation, which could remove some of the water-soluble non-starch polysaccharides, cotton phenols, tannins, phytic acid, and other anti-nutritional factors, minimize the degree of protein heat denaturation, and enhance the protein content and nutritional value [7,13].Numerous studies have shown that CPC acts as a novel protein source with different effects on growth performance and immunity and intestinal flora of different aquatic animals [14,15]. In studies on golden pompano (*Trachinotus ovatus*) and rainbow trout (*Oncorhynchus mykiss*), it was found that replacing moderate amounts of fishmeal with CPC did not negatively affect the growth performance and immune response [10,16]. Similar studies have been reported in ♀*Epinephelus Fuscoguttatus* × ♂*Epinephelus Lanceolatu* via [17]. In addition, Wang et al. showed that replacing moderate amounts of fishmeal with CPC improved the growth performance, immunological response, and digestibility of *L. vannamei* [18]. However, excessive replacement of fishmeal with CPC will adversely affect the intestinal health of aquatic animals, causing intestinal inflammation and affecting immune enzyme activity, thus affecting growth performance [14,19]. In addition, Yin et al. showed that excessive replacement of fishmeal with CPC resulted in ♀*E. Fuscoguttatus* × ♂*E. Lanceolatu* via of liver in grouper, thus reducing the antioxidant capacity [15]. A similar study found that excessive replacement of fishmeal with CPC negatively affected the growth, body composition, hemolymph indicators, and blood enzyme activity of *L. vannamei* [20]. Overall, the mechanism of CPC as a novel plant protein source to replace fishmeal in *L. vannamei* has been less studied up to now.

High-throughput sequencing has become a routine experimental technique used in the field of life science [21]. Transcriptome sequencing (RNA-SEQ) is a new technology that uses high-throughput sequencing to study gene transcription, which can comprehensively and quickly obtain the sequence and expression information of almost all transcripts of a specific cell or tissue in a certain state [22,23]. It can accurately analyze gene expression differences, structural variation, molecular markers, and other important problems in life science. High-throughput sequencing has also been widely used in transcriptome analysis to study the mechanisms of growth, development, immunity, and other related life activities. Hou et al. used transcriptome sequencing analysis to find that dietary TWS119 may improve growth performance and immune resistance in *L. vannamei* by activating the Wnt/β-catenin pathway and inhibiting the activity of LvGSK3β [21]. Yin et al. employed transcriptome sequencing analysis to discover that lipids from *L. vannamei* are a source of nutrition for *Vibrio vulnificus*, and they are identified by regulating lipid homeostasis to avoid being infected [22].

Although our previous study showed that moderate amounts of CPC in place of fishmeal can improve the growth performance of *L. vannamei* [18], there have been limited investigations of the mechanism of the immunological response to CPC substitution for fishmeal. In this study, high-throughput sequencing analysis was employed to investigate the changes at gene transcription level, aiming to clarify the mechanism by which CPC substitution for fishmeal affects the immunological response in *L. vannamei*. The results can provide a theoretical basis for the shrimp feed industry, as it seeks new protein sources to replace fishmeal.

## 2. Materials and Method

### 2.1. Experimental Diets

Four isonitrogenous and isoenergetic diets containing different levels of CPC are shown in Table 1. Fishmeal, soybean meal, and peanut meal were used as intact protein sources; fish oil and lecithin were used as the lipid sources; and flour was used as the carbohydrate source. The groups were recorded as FM (0 replacement level, 0%), CPC15 (low replacement level, 15%), CPC30 (middle replacement level, 30%), and CPC45 (high replacement level, 45%). The ingredients were ground into a fine powder, sieved through an 80-mesh size, and precisely weighed according to the formula. The micro constituents were mixed homogenously by the sequential expansion method. Then, the lipids and deionized water were added and thoroughly mixed to obtain a homogenous mixture. Subsequently, the dough was passed through the pelletizer with 1.0 mm and 1.5 mm diameters and dried at 60 °C in a ventilated oven for 0.5 h. Dry pellets were placed in plastic bags and stored at −20 °C until needed.

CPC products were provided by Xinjiang Jinlan Co., Ltd. (Xinjiang, China). The amino acid composition of the experimental diets is shown in Table 2.

### 2.2. Feeding Trial and Experimental Conditions

*L. vannamei* were purchased from Hengxing 863 Fisheries Science and Technology and temporarily reared in a 4.5 m × 4.9 m × 1.8 m outdoor cement pond for two weeks to adapt to the test conditions. The experiment was conducted in a fiberglass tank (0.3 m^3^). Healthy and strong individuals, with an average body mass of 0.42 ± 0.01 g and without visible injuries, were selected and divided into four groups of three replicates, each with 40 shrimps per replicate and cultured for eight weeks. Shrimps were fed the experimental diets to apparent satiation four times daily (07:00, 11:00, 17:00, and 21:00) for eight weeks. During the experimental period, water quality parameters were measured daily at 7:00 and 21:00 to ensure the temperature was 29.0–30.0 °C, the salinity was 27–30 g∙L^−1^, the dissolved oxygen level was at least 6.0 mg∙L^−1^, the pH value was 7.7–8.0, and the ammonia nitrogen level was lower than 0.05 mg∙L^−1^.

### 2.3. Sample Collection

This study protocol was approved by the ethics review board of Guangdong Ocean University. All procedures were performed in accordance with the Declaration of Helsinki and relevant policies in China. At the end of the eight-week period, shrimp were fasted for 24 h before collecting samples. Hemolymph collected from three shrimp (per tank) was used as a sample, which was centrifuged (4000× *g*) at 4 °C for 15 min after storage at 4 °C for 12 h, and the serum supernatant was obtained for the analysis of enzyme activeties. Hemolymph from another three shrimp randomly selected from each tank was withdrawn into modified ACD anticoagulant solution and centrifuged for 5 min (3000× *g* at 4 °C) to isolate hemocytes, using as a sample. The hemocytes were stored at −80 °C for RNA extraction and used for the detection of gene expression and transcriptomic analysis, as our previous study showed [21].

### 2.4. Evaluation of Non-Specific Immune Indices

The phosphatase (ACP), alkaline phosphatase (AKP), superoxide dismutase (SOD), phenol oxidase (PO), lysozyme (LZM), hydrogen peroxidase (CAT), alanine aminotransferase (ALT), aspartate transaminase (AST), and malondialdehyde (MDA) were measured using the ACP and AKP assay kit (Cat. No. A060-2), SOD assay kit (WST-1 method) (Cat. No. A001-3), PO kit (Cat. No. H247), LZM assay kit (Cat. No. A059-2), CAT assay kit (visible light) (Cat. No. A007-1), ALT assay kit (Cat. No. A022-2), and AST kit (Cat. No. A247), respectively. The content of MDA was determined using an MDA assay kit (TBA method) (Cat. No. A003-1). All these assay kits were purchased from the Nanjing Jian Cheng Bioengineering Institute (Nanjing, China) and used as per the manufacturer’s instructions.

### 2.5. Gene Expression Analysis

Total RNA was extracted from hemocytes samples using TransZol Up Plus RNA kits (TransGen, China) following the manufacturer’s protocol, and the quality and concentration were assessed by spectrophotometric analysis (Nanodrop 2000). Total RNA (1 μg) was used for cDNA synthesis by PrimeScript^TM^ RT reagent kit with gDNA Eraser (AG, China) according to the manufacturer’s instructions. Using qPCR, the gene expressions of *PO*, copper superoxide dismutase (*SOD1*), manganese superoxide dismutase (*SOD2*), *CAT*, *LZM*, *AKP*, *ACP*, *ALT,* and *AST* were determined. The real-time PCR for the target genes was performed on a Light Cycler 480 with a SYBR^®^ Green Premix Pro Taq HS qPCR Kit II by the following program: 1 cycle at 95 °C for 30 s, 40 cycles at 95 °C for 5 s, 57 °C for 30 s, and 78 °C for 5 s. Elongation factor 1α (EF1α, GenBank accession No. GU136229) was used as the internal control. Three replicated qPCRs were performed per sample. The primer sequences are listed in Table 3.

### 2.6. Transcriptome Sequencing and Analysis

#### 2.6.1. RNA Extraction and Transcriptome Sequencing

Total RNA of hemolymph of *L. vannamei* from FM, CPC15, CPC30, and CPC45 groups was extracted by the Trizol method. The extracted RNA was treated with RNase-free DNase I to avoid residual genomic DNA contamination. The quality of the isolated RNA was checked by gel electrophoresis on a 1.5% agarose gel and NanoDrop 200 spectrophotometer, USA, and 5 µg total RNA was collected for each group. The library was constructed using the Illumina TruSeq RNA Sample Preparation Kit (Illumina, USA). The mRNA was enriched by magnetic beads, which contained Oligo (dT) (Tiangen, China). Library construction and sequencing were carried out by Guangzhou Gene Denovo Biotech Co., Ltd. (Guangzhou, China). Briefly, the mRNA was enriched with magnetic beads with Oligo (dT), and the fragmentation buffer was added to break the mRNA into small fragments. Using the post-fragment mRNA as a template, random hexamers were used to synthesis the first strand of cDNA. dNTPs, buffer, DNA polymerase I, and RNase H were then used to synthesise the second strand of cDNA using Agencourt AMPure XP Beads reagent. The cassette was purified, and EB buffer eluted with end-repair, base A, and sequencing linker, and the target fragment was recovered by agarose gel electrophoresis. Finally, PCR was used to complete the entire library preparation work. The PCR product was purified to create the final cDNA libraries. Lastly, the library preparations were sequenced on an Illumina HiSeqTM platform that generated ~200 bp pair–end raw reads.

#### 2.6.2. Data Analysis

Before the filtered reads were mapped to the *L. vannamei* reference genome (NCBI Genome database ID: No. PRJNA438566), using HISAT software [21], raw reads generated by Illumina Hiseq 2000 from the twelve libraries were cleaned using SeqPrep (https://github.com/jstjohn/SeqPrep; accessed on 1 October 2021) and Sickle (https://github.com/najoshi/sickle; accessed on 1 October 2021) software by removing reads with adaptors, reads with more than 10% Q < 20 bases (those with a base quality less than 20), and low-quality sequences (reads with ambiguous bases ‘N’).2.4.3. Serum Biochemical Parameters

#### 2.6.3. Differential Expression Analysis and Functional Annotation

As shown in a previous report, the abundance of all genes was normalized and calculated using uniquely mapped reads by RPKM (read per kilobase of exon model per million mapped reads) [23]. Log2(FC) was used as an indicator of the transcriptomic differences among the FM, CPC15, CPC30, and CPC45 groups. A false discovery rate (FDR) <0.001 was used as the threshold of the *p*-value in multiple tests to determine the significance of gene expression differences. Genes were considered differentially expressed when the FDR ≤ 0.001 and a greater than twofold change (absolute value of log2 ratio > 1) in expression across libraries was observed. The uniqueness was DEGs if the absolute value of log2(FC) was greater than 1 and the FDR was less than 0.05. DEGs were analyzed using tools in the GO (http://www.geneontology.org/ accessed on 1 October 2021) and KEGG (https://www.genome.jp/kegg accessed on 1 October 2021) databases. The Series Test of Cluster was performed using STEM software with log2 normalization for gene expression preprocessing. The *p*-value was used to measure the number of genes within the module in relation to the expected value of the random distribution. The lesser the *p*-value, the more significant the gene set.

### 2.7. Validation of DEGs by qPCR

To validate the RNA-Seq data, RNA samples for transcriptomes were measured using qPCR. Sixteen DEGs and eight AMP genes were selected, and cDNA samples were prepared from transcriptome-sequenced samples using the PrimeScript™ RT kit and the gDNA Eraser (Perfect Real Time) kit (Takara, Japan). Primers were designed using Primer Premier 5 software (Table 3), and their relative expression was quantified using the qPCR method described above.

### 2.8. Statistical Analysis

All results were subjected to one-way analysis of variance (ANOVA) followed by Duncan’s multiple range test to determine significant differences among treatment groups. Statistical analysis was performed using SPSS version 22.0 (SPSS Inc, Chicago, IL, USA), and all data were expressed as mean ± standard deviation (SD), with *p* < 0.05 indicating significant differences.

## 3. Results

### 3.1. Non-Specific Immune Indices in the Serum

Non-specific immune indices in the serum are shown in Table 4. The activities of ACP and LZM in the serum of *L. vannamei* showed a trend of increasing and then decreasing with the increase of the CPC substitution ratio. Moreover, the values of ACP and LZM were significantly higher in the CPC30 group than in the FM group. (*p* < 0.05). In addition, the values of PO were not significantly different among the groups. However, the values of AKP, SOD, and CAT were significantly higher in the CPC15, CPC30, and CPC45 groups than in the FM group. (*p* < 0.05). Further, the AST and ALT values in the CPC30 group were significantly lower than those in the FM group (*p* < 0.05). In addition, the levels of MDA were considerably lower in all the CPC-added groups than in the FM group (*p* < 0.05).

### 3.2. Expression of Non-Specific Immune Genes in Hemocytes

Overall, the expression of immune genes in the hemocytes could be induced to varying degrees in the CPC-containing group compared with the FM group, and the expression of some genes was significantly higher in several groups (Figure 1). Specifically, the expression of *PO* and *LZM* was substantially higher in the CPC30 group compared with the FM group, and there was no significant difference in the expressions of the remaining groups (*p* < 0.05). However, the *ACP* expression in the CPC45 group was significantly lower than that in the FM group (*p* < 0.05). Gene expressions of *SOD1*, *AKP*, and *AST* were significantly higher in most CPC-containing groups than in the FM group (*p* < 0.05). The highest expression of *AKP* and AST was found in the CPC45 group, while *SOD1* expression was highest in the CPC15 group (*p* < 0.05). The expressions of *SOD2* and *ACT* peaked in the CPC15 group compared with the FM group and were significantly higher than in the rest of the groups (*p* < 0.05). In addition, there was no significant difference in the expression of *ALT* among the groups.

### 3.3. Assembly and Sequence Alignment Analysis

The raw data of this study have been deposited in the SRA database with the accession number PRJNA808042. Four cDNA libraries from the mRNA extracted from the hemocytes of the *L.vannamei* in FM, CPC15, CPC30, and CPC45 groups were sequenced. Sequencing was performed using the Illumina platform, and the total number of clean data averaged 85,783,958,047 (bp) after removing articulators and filtering low-quality sequences, as shown in Table 5. Filtering from the raw reads yielded an average of 576,681,882 clean reads. Among them, the FM group generated 140,046,034 high-quality average reads from 140,467,610 raw average reads, and the CPC15 group generated 139,776,820 high-quality average reads from 140,162,764 average raw reads, with Q20% greater than 97% and Q30% greater than 92% of the mean for each group. There were 546,587,036 valid data for the total mean, and the mapping rate was greater than 62% when comparing the analysis statistics with the reference genome data. The assembly results indicate that the sequencing was of good quality and could be used for transcriptome analysis.

### 3.4. Identification of DEGs

Compared with *L. vannamei* in the FM group, 769 (448 up, 321 down), 649 (266 up, 383 down), and 902 (434 up, 468 down) DEGs were identified in the CPC15, CPC30, and CPC45 groups, respectively (Figure 2). As shown in Figure 3, 121 of these DEGs are shared. In addition, CPC15, CPC30, and CPC45 had 367, 295, and 521 unique DEGs, respectively, compared with the FM group.

### 3.5. GO Enrichment Analysis of the DEGs

All DEGs were mapped to terms in the GO database to evaluate their functions. As shown in Figure 4, 51 significantly (*p* < 0.05) enriched terms were identified and classified into three major functional classes, including biological processes, molecular functions, and cellular components. There were 24, 10, and 17 significantly enriched terms classified into biological processes, molecular functions, and cellular components, respectively. In the category of biological processes, DEGs are annotated mainly to single-organism processes (109 up, 106 down), cellular processes (120 up, 100 down), and metabolic processes (104 up, 80 down). Among the molecular functions, catalytic activity and binding dominate (112 up, 75 down). In addition, the highest content of cellular fractions was classified as cell (130 up, 83 down) and cell parts (130 up, 83 down).

### 3.6. KEGG Enrichment Analysis of the DEGs

On the basis of KEGG enrichment analysis, all DEGs were annotated to cellular processes, environmental information processing, genetic information processing, metabolism, human diseases, and Organismal Systems. Moreover, these annotated DEGs were further classified into 42 subclasses (Figure 5). Consistently, the largest group in these subcategories is the global overview map. However, the top 20 pathways were not identical among the CPC-contained groups and FM group. As shown in Figure 6, only the top 20 pathways of CPC45 compared with the FM group were functionally associated with five KEGG classes, with the top five signalling pathways being aldosterone-regulated sodium reabsorption (ko04960), platinum drug resistance (ko01524), toxoplasmosis (ko05145), small cell lung cancer (ko05222), and pathogenic *Escherichia coli* infection (ko05130; Figure 6C). Among the top 20 signalling pathways that were significantly enriched in KEGG in the CPC15 group compared with the FM group, seven were related to Organismal Systems, four to metabolism and six to human diseases, with the top five signalling pathways being IL-17 signalling pathway (ko04657), Toll and immune deficiency (IMD) pathways (ko04624), apoptosis (ko04210), carbohydrate digestion and absorption (ko04973), and beta-Alanine metabolism (ko00410; Figure 6A). However, nine of the top 20 pathways significantly enriched in KEGG in the CPC30 group were related to body systems, three to metabolism, and five to human diseases. The first five of these signalling pathways were caffeine metabolism (ko00232), platinum drug resistance (ko01524), steroid biosynthesis (ko00100), Th17 cell differentiation (ko04659), and toxoplasmosis (ko05145; Figure 6B). In addition, compared with the FM group, analysis of DEGs containing the CPC group revealed that among the common differential pathways, platinum drug resistance, toxoplasmosis, pathogenic *Escherichia coli* infection, steroid biosynthesis, measles, carbohydrate digestion and absorption, insulin resistance, and Toll and IMD pathways were significantly enriched (*p* < 0.05; Figure 6). However, among the 20 most affected signalling pathways, the NF-kappa-B inhibitor cactus (Cactus) gene is of particular interest because of its involvement in several different signalling pathways, including the toxoplasmosis, pathogenic *Escherichia coli* infection, insulin resistance, and Toll and IMD pathway (Table 6).

### 3.7. Series Test of Cluster

Series Test of Cluster is shown in Figure 7, and there are four significant trends, with 54, 32, 14, and 9 DEGs, respectively. In addition, Cactus gene expression decreased and then increased with an increasing substitution ratio (Figure 7B). Cactus gene expression was considerably lower in all CPC-containing groups than in the FM group, with the lowest value observed in the CPC30 group (*p* < 0.05; Figure 7C).

### 3.8. Validation of qPCR

Sixteen DEGs were randomly selected for validation, including seven up-regulated expression genes and nine down-regulated expression genes. The qPCR results showed the same expression trend as the high-throughput sequencing data. Thus, the qPCR analysis results confirmed the expression of DEGs detected in the high-throughput sequencing analysis (Figure 8).

### 3.9. Expression of AMP Genes in Hemocytes

As shown in Figure 9, the expression of *Pen2*, *Pen3*, *Pen4*, and *Cru3* genes was significantly higher in the group with CPC with the diet than in the FM group (*p* < 0.05). In addition, the expression of *Cru1*, *ALF1*, and *ALF3* genes showed a trend of increasing and then decreasing with the increase in the CPC substitution ratio, and the expression in the CPC30 group was substantially higher than that of the FM group (*p* < 0.05). However, the expression of the *ALF2* gene peaked in the CPC45 group and was significantly higher than that in the FM group (*p* < 0.05).

## 4. Discussion

A series of studies have confirmed that replacing fishmeal with a certain amount of CPC can improve the immune capacity of aquatic animals [15,24]. However, there have been fewer studies on how CPC improves immune mechanisms in animals. In addition, studies using high-throughput sequencing to assess CPC on gene transcription levels are uncommon. In this study, CPC influenced the immune response of *L. vannamei*, and the immune response was studied by detecting immune enzymes and analyzing gene expression and transcriptome sequencing. These studies indicate that CPC increased non-specific immune enzyme activity and gene expression in *L. vannamei* and may induce the expression of AMP genes by regulating Cactus genes, ultimately improving the immunity of *L. vannamei*.

Dietary sources of plant protein have been shown to influence growth performance and innate immune responses in animals [25]. As a type of plant protein, CPC has been proved to improve the growth performance of *L. vannamei* when used as a replacement of fishmeal [18]. *L. vannamei* is an invertebrate that relies on its innate immune system to combat disease [18,26,27,28]. Therefore, various non-specific immune indices are often chosen to assess the impact on immunity [29]. Commonly used indices include enzyme activities such as ACP, AKP, SOD, PO, LZM, CAT, and the level of MDA. To determine the effect of the CPC substitution of fishmeal on the immunity of *L. vannamei*, we analyzed non-specific immunity indicators. In the present study, the activities of AKP, SOD, and CAT were significantly higher in all CPC-containing groups, while the activities of MDA were considerably lower in all CPC-containing groups (*p* < 0.05). A similar study showed that replacing 36% FM with CPC significantly increased SOD and GSH-PX activities while decreasing MDA levels [10]. In addition, it has also been shown that replacing no more than 24% of fishmeal with CPC can significantly increase the activity of SOD and CAT and decrease the MDA content of groupers [20]. In shrimp, LZM is an important immune enzyme that can lyse microbial cell walls and kill microorganisms and is a direct indicator of immune competence [30]. Ye et al. showed that replacing up to 20% of fishmeal with CPC increased LZM activity in grouper, while replacing no more than 60% resulted in no significant change in LZM activity [15]. In addition, it has been suggested that replacing 45% of fishmeal with CPC can improve the activity of LZM [18]. Consistently, LZM activity was significantly higher in the CPC30 diet group than in the FM group in this study, with no significant difference among the remaining groups. This indicates that replacing fishmeal with a moderate amount of CPC increases LZM activity. ALT and AST are the most important transaminases involved in amino acid metabolism and are direct indicators of health [31]. In the present study, the activity of AST showed an increasing trend with an increasing substitution ratio, and the activity of AST and ALT was significantly higher in the CPC45 group than in the other groups. Similar studies on *L. vannamei* showed an increasing trend in ALT and AST activities as the proportion of CPC substituted fishmeal increased [20]. In addition, studies have pointed out that many enzymes are dimeric or tetrameric and that each of these subunits is determined by a gene [32,33]. Thus, it is not entirely accurate that the expression level of a gene is consistent with the activity of the corresponding enzyme. The expression levels of PO enzyme and *PO* gene were not completely consistent in this study. This may be a result of the enzyme activity being influenced by a number of factors. Nonetheless, CPC could have immune-boosting effects in *L. vannamei*, as evidenced by gene expression of several immunological enzymes. The probable reason is that CPC is a high-quality plant protein source with increased content of essential amino acids, such as Arginine. In addition, reasonable processing has reduced the degree of thermal denaturation of its protein and made it easier for animals to digest and absorb [13].

Transcriptome sequencing, a common tool in biology research at present, is also extensively used to analyses the mechanisms associated with protein source substitution in diets on the growth, metabolism and immunity of organisms [21,34,35]. In the present study, a comparative transcriptome analysis of *L. vannamei* containing the CPC and FM groups was performed to investigate the mechanism of the effect of the CPC substitution for fishmeal. GO and KEGG analysis showed that the distribution of DEG enrichment functions and the classification of DEG-related signaling pathways were essentially the same among the different CPC-containing and FM groups. Therefore, it can be seen that the effects of different levels of CPC-substituted fishmeal on *L. vannamei* at the transcriptome level are approximately the same. In the DEG enrichment analysis, it was found that Cactus genes affect many different pathways of interest. Examples include toxoplasmosis, pathogenic *Escherichia coli* infection, insulin resistance, and Toll and IMD pathway. These pathways are all related to the immune pathway, suggesting that CPC substitution for fishmeal affects the immune response of shrimp. The NF-κB pathway, as is well known, is often considered the main regulatory pathway of the shrimp immune response [36]. Bacterial disease infections can be recognized by Toll and IMD pathways, which then activate two NF-κB transcription factors, Dorsal and Relish, respectively, ultimately promoting the expression of various AMP genes [37,38,39]. Cactus functions as an inhibitor of Dorsal and can negatively regulate the NF-κB pathway [40,41]. It contains an N-terminal regulatory region responsible for ubiquitin recognition and proteasomal degradation and has an adjacent Ankyrin repeat sequence that is capable of binding to the Rel-homology region [42]. The Rel-homology region and a destabilizing C-terminal PEST domain are required to inhibit DNA binding [42]. In the present study, Cactus expression was considerably decreased in the CPC-containing group compared with the FM group, reaching its lowest point in the CPC30 group. Consistently, the expression of AMP genes was also differentially induced to elevated levels in the CPC-containing group of the diet. In conclusion, the CPC-containing group in the diet may maintain the normal regulation of the NF-κB pathway in animals by regulating the expression of the Cactus gene, promoting the expression of various AMP genes, and ultimately improving the disease-fighting immunity of *L. vannamei*. This was further verified by the gene expression of these AMP genes. However, the exact mechanism needs further validation.

## 5. Conclusions

In the present study, dietary moderate amounts of CPC significantly improved the non-specific immunize activity and expression of *L. vannamei*, which may be due to the fact that CPC increases the expression of AMP genes by suppressing the expression of Cactus genes, thereby possibly resulting in ultimately improving the immunity of *L. vannamei*. The results of this study contribute to understanding the mechanism of CPC as an alternative to fishmeal and provide useful information for the development of novel protein sources in shrimp feed.

## Figures and Tables

**Figure 1 animals-13-01185-f001:**
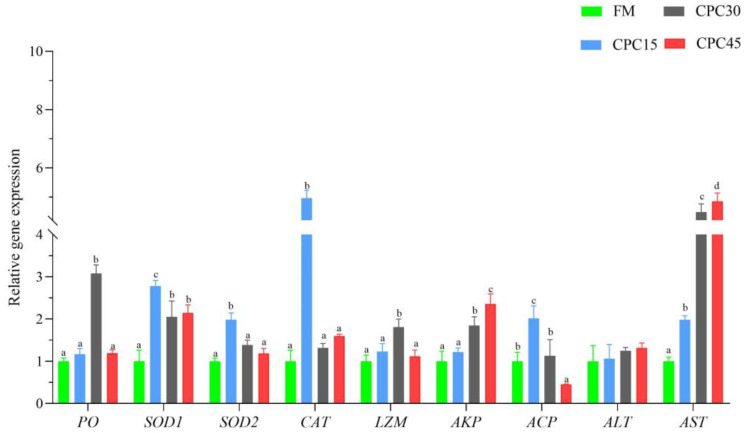
The expression of immune genes in the hemocytes of *L. vannamei* fed with increasing levels of CPC. Different superscript letters indicate significant differences exist among treatments (*p* < 0.05).

**Figure 2 animals-13-01185-f002:**
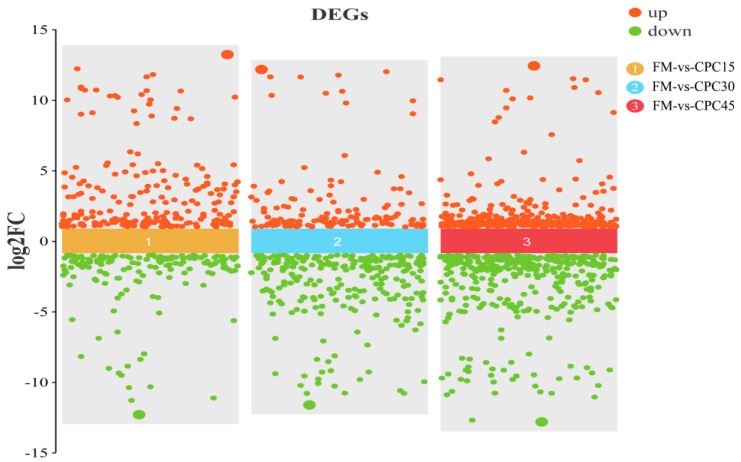
Differentially expressed genes (DEGs) between *L. vannamei* containing CPC diet and FM groups.

**Figure 3 animals-13-01185-f003:**
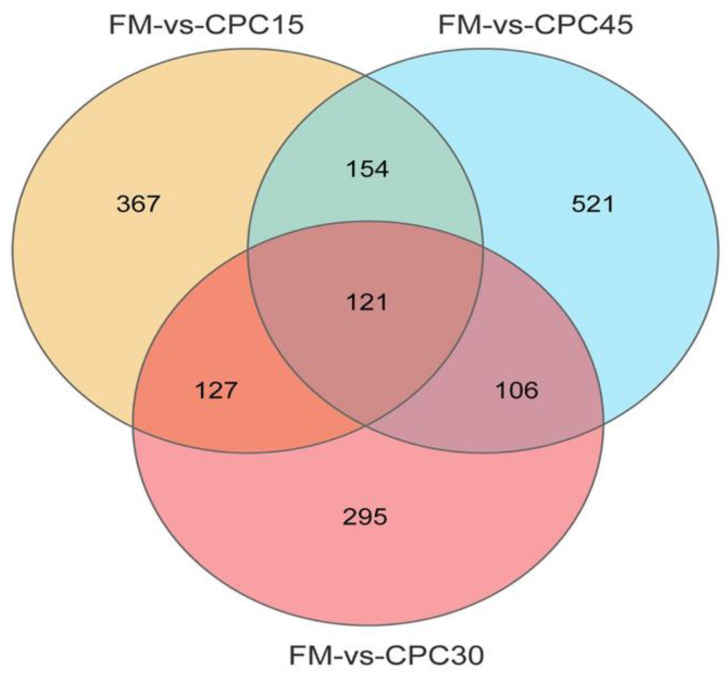
Venn diagram of DEGs.

**Figure 4 animals-13-01185-f004:**
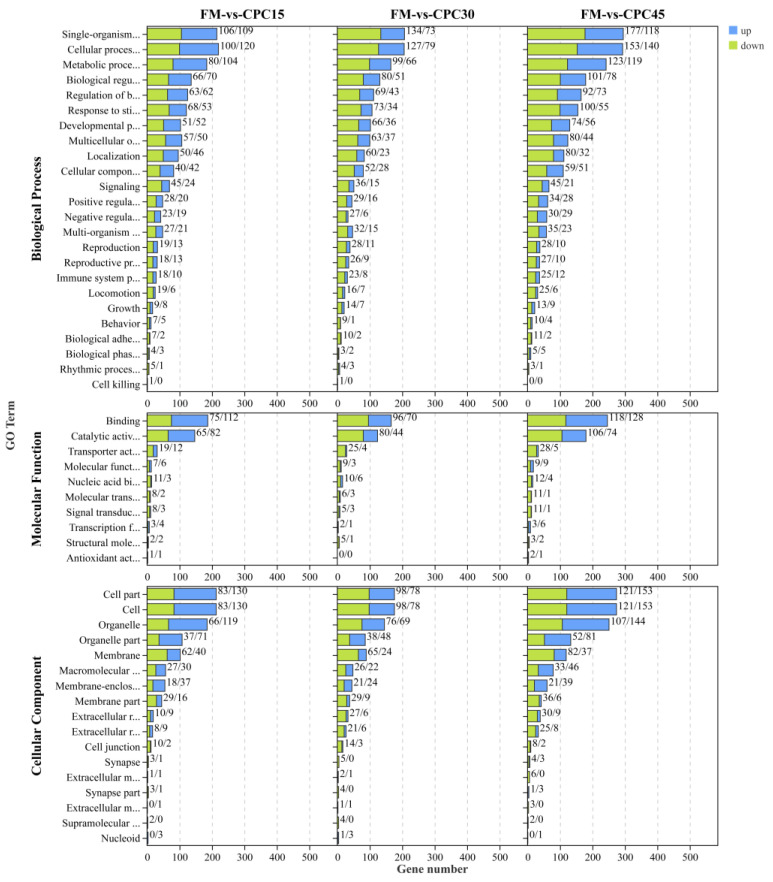
GO enrichment analysis of DEGs containing CPC-fed *L. vannamei* and FM groups. Three main GO categories: cellular components, molecular functions, and biological processes. *x*-axis indicates the number of DEGs, and *y*-axis indicates GO categories and subcategories.

**Figure 5 animals-13-01185-f005:**
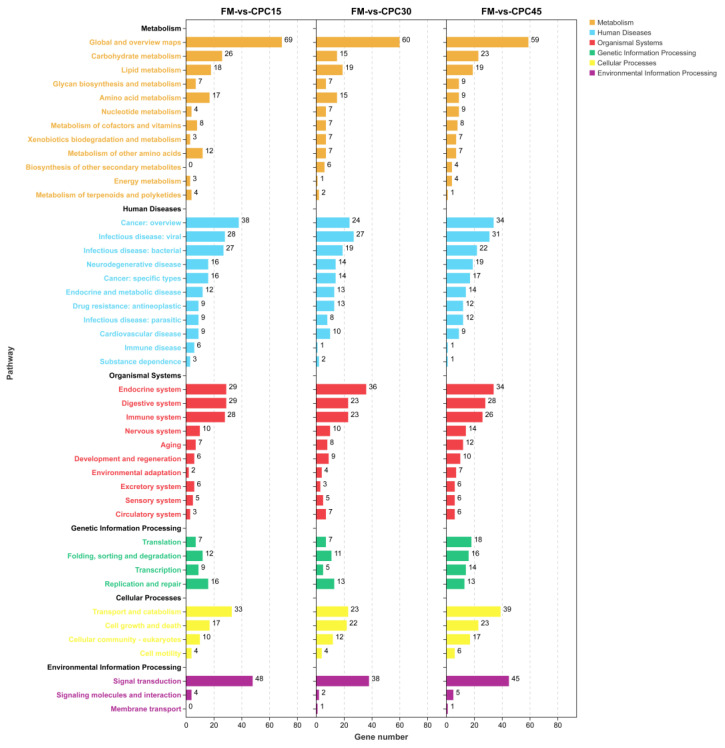
KEGG enrichment analysis of DEGs containing CPC-fed *L. vannamei* and FM groups. Highly expressed biological pathways in the transcriptome were retrieved from the KEGG database. DEGs were assigned to six specific KEGG pathways, including organismal systems, metabolism, genetic information processing and environmental information processing, cellular processes, and human diseases.

**Figure 6 animals-13-01185-f006:**
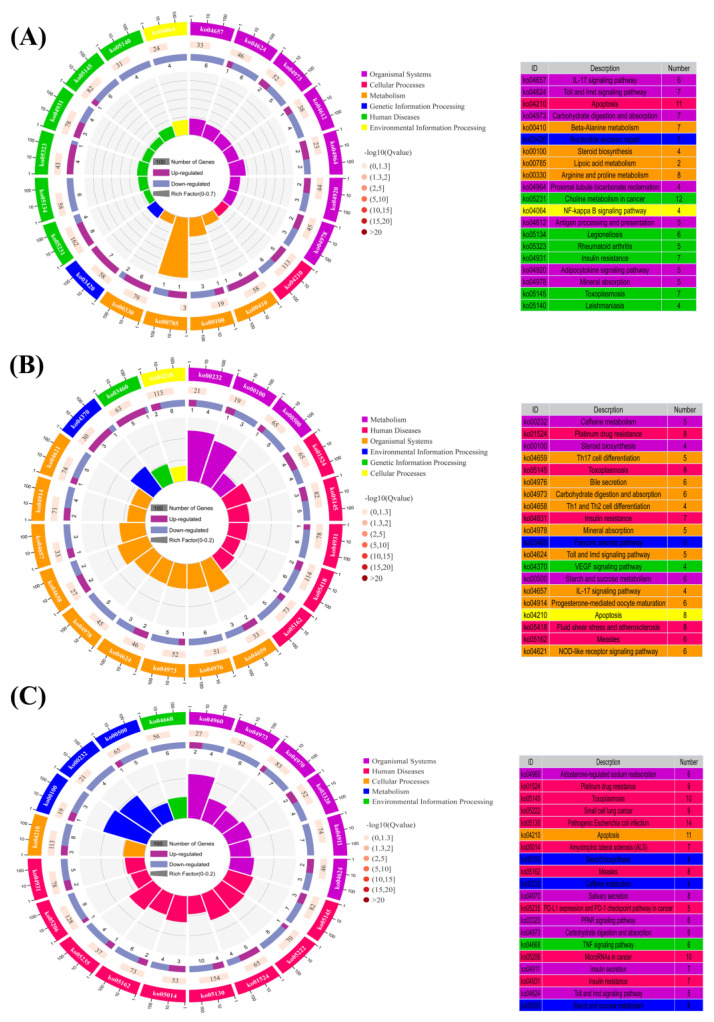
Top 20 pathways for KEGG enrichment analysis. (**A**) FM group vs. CPC15 group; (**B**) FM group vs. CPC30 group; and (**C**) FM group vs. CPC45 group. The first circle indicates the top 20 pathways enriched from outside to inside, with a coordinated scale of gene numbers outside, and the number and color of the number on the first circle are the same as the number and color of the number on the right side of the chart. The second circle indicates the number of genes in the background and the *p*-value of each pathway. The more genes, the longer the bar. The smaller the *p*-value, the darker the color. The third circle indicates the ratio of up-regulated genes (dark purple) to down-regulated genes (light purple), with the numbers inside the bar. The fourth circle indicates the enrichment factor (the proportion of DEGs in the background genes) for each pathway. Each cell of the background auxiliary line represents 0.1. The different colors of the first and fourth circles represent different pathway classifications.

**Figure 7 animals-13-01185-f007:**
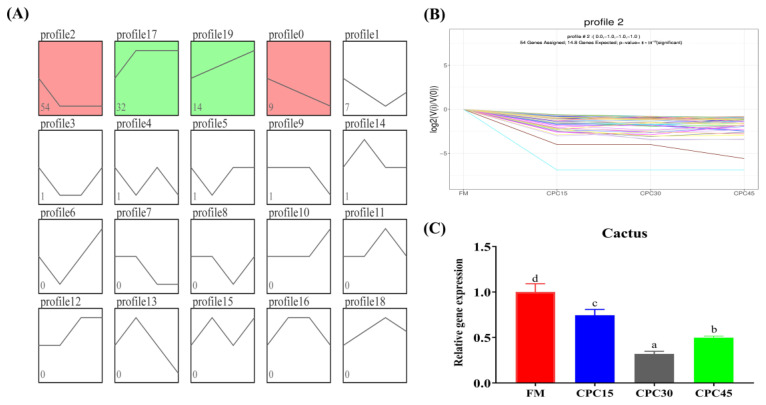
Trend analysis of shared differential genes. (**A**) analysis of shared gene trends. The difference is not significant for the white background and significant for the rest of the color background (*p* < 0.05); (**B**) analysis of all profile2 trends; and (**C**) expression of Cactus genes in profile2 trends. Different letters indicate significant differences (*p* < 0.05).

**Figure 8 animals-13-01185-f008:**
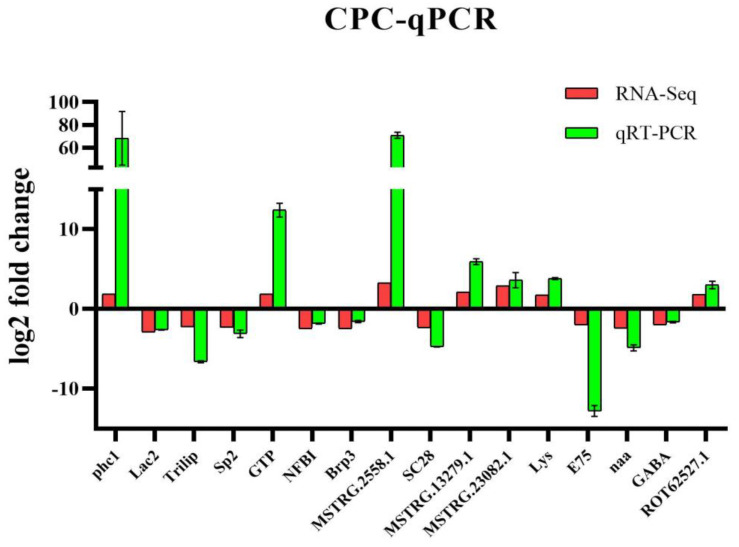
Validation of the associated DEGs by qPCR. Gene expression assays were performed in triplicate for each sample. Expression values were normalized to those of EF1α using the Litvak (2^−ΔΔCt^) method, and data are the mean ± SD of triplicate assays.

**Figure 9 animals-13-01185-f009:**
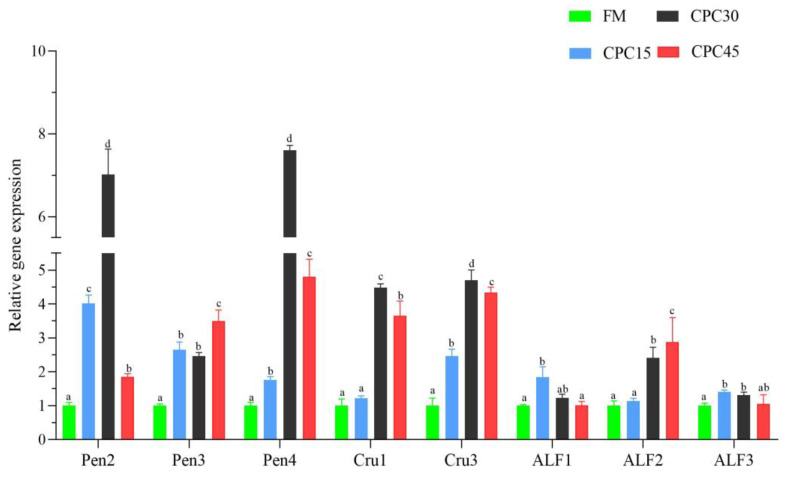
The expression of antimicrobial peptide genes in *L. vannamei* by different levels of CPC-substituted fishmeal. Different superscript letters indicate significant differences exist among treatments (*p* < 0.05).

**Table 1 animals-13-01185-t001:** Formulation and proximate composition of the experimental diets (Air dry basis, %).

Ingredient	Group			
	FM	CPC15	CPC30	CPC45
Fishmeal	25	21.25	17.5	13.75
Soybean meal	25	25	25	25
Peanut meal	8.92	8.92	8.92	8.92
Flour	18	18	18	18
Brewer’s yeast	2.84	2.84	2.84	2.84
Shrimp head powder	5.53	5.53	5.53	5.53
Cottonseed protein concentrate	0	4.39	8.77	13.16
Fish oil	2	2	2	2
Soybean oil	2.21	2.46	2.72	2.98
Choline	0.3	0.3	0.3	0.3
Soy lecithin	1	1	1	1
Vitamin and mineral premix ^a^	1	1	1	1
Choline chloride	1.5	1.5	1.5	1.5
Vitamin C	0.1	0.1	0.1	0.1
Microcrystalline cellulose	6.55	5.66	4.77	3.87
Antioxidant	0.05	0.05	0.05	0.05
Total	100	100	100	100
Proximate composition				
Crude protein	38.11	38.47	38.21	38.37
Ether extract	6.74	6.73	6.78	6.76
Moisture	7.52	7.61	7.59	7.46
Ash	10.87	11.80	11.48	10.80

^a^ Vitamin and Mineral premix (per kg diet): Vitamin A, 100,000 IU; Vitamin D3, 10,000 IU; Vitamin E, 4000 mg; Vitamin K3, 1000 mg; Vitamin B1, 500 mg; Vitamin B2, 1000 mg; Vitamin B6, 1000 mg; Vitamin B12, 2.0 mg; Vitamin C, 15,000 mg; Nicotinic acid, 4000 mg; Calcium pantothenate, 2000 mg; Folic acid, 100 mg; Biotin, 10.0 mg; Iron, 10,000 mg; Copper, 300 mg; Zinc, 5000 mg; Manganese, 1200 mg; Iodine, 80 mg; Selenium, 30 mg; and Cobalt, 20 mg.

**Table 2 animals-13-01185-t002:** Amino acid composition of the feed (dry matter basis, %).

Item	FM	CPC15	CPC30	CPC45
Asp	3.51	3.53	3.6	3.71
Thr	1.48	1.43	1.44	1.34
Ser	1.65	1.59	1.64	1.64
Glu	6.45	6.68	6.94	7.1
Gly	1.92	1.89	1.84	1.88
Ala	1.87	1.83	1.76	1.83
Cys	0.42	0.45	0.43	0.45
Val	1.63	1.65	1.64	1.61
Met	0.59	0.57	0.54	0.46
Ile	1.44	1.45	1.44	1.45
Leu	2.59	2.58	2.57	2.57
Tyr	0.98	0.98	1.06	0.96
Phe	1.64	1.72	1.78	1.53
Lys	2.29	2.2	2.14	2.12
His	0.97	0.97	1.02	1.11
Arg	2.38	2.57	2.79	3.12
Pro	1.82	1.82	1.86	1.77
Total amino acids	33.63	33.91	34.49	34.65

Note: total amino acids, including threonine, valine, methionine, isoleucine, leucine, phenylalanine, lysine, histidine, arginine, serine, glutamic acid, glycine, alanine, tyrosine, and proline.

**Table 3 animals-13-01185-t003:** Primers used for qPCR.

Gene ID No.	Primer Name	Sequence (5′-3′)
GU136229	EF1α-F	GAAGTAGCCGCCCTGGTTG
	EF1α-R	CGGTTAGCCTTGGGGTTGAG
DQ005531	SOD-F	CTTTGCCACCCCTCAAGTATG
	SOD-R	TGCCTCCGCCTCAACCA
XM027381766	PO-F	AAGCCAGGCAGCAACCAC
	PO-R	CAGAAGTTGAAACCCGTGGC
AF425673	LZM-F	TATTCTGCCTGGGTGGCTTAC
	LZM-R	CAGAGTTGGAACCGTGAGACC
JX162772	CAT-F	ACAAGACGGACCAAGGCATC
	CAT-R	ACTTATATCCTCAGTGACTGGCATG
KR676449	ACP-F	TCAAGACCTGGAAGAATGGGACT
	ACP-R	CCTCTCTGAACCTCCTCCTGTAAC
XM027353927	AKP-F	TACGGCTACCGAAGGGACG
	AKP-R	AGTCGGTGTTGTTCGTGTTGG
XM027365017	AST-F	AGATGGACCAGCGTGTGATTAC
	AST-R	TCGCCTTGGTCACGCTGT
XM027364929	ALT-F	TCTCCGCATCGGCTCCA
	ALT-R	CCACAGGTCTTTGGGTCGTAG
ROT83547.1	phc1-F	CGAAGGTCGCTGGAATGC
	phc1-R	ATTCCGTGGACTTGCGTTC
ROT81694.1	Lac2-F	CAGATGGTAACCCCGTTGTGC
	Lac2-R	GAAGGTTCCGACTTTGGTGTTTT
ROT80984.1	Trilip-F	CGGCTTTATGGAGGGGATG
	Trilip-R	ACTCGTTCGTCACAGGCATAAG
ROT69062.1	Sp2-F	GCGTGGAAGGGAAGTGGG
	Sp2-R	TTCGGCTCACGGGCTTCT
ROT66027.1	GTP-F	TGCTGTCCAACACCATCTTCAC
	GTP-R	ACATCGTTCGCCGTTTCCT
ROT63397.1	NFBI-F	ACTCCTTGCCGCTGACCTTAC
	NFBI-R	GTGCCGTCCGACCACTCTT
ROT85909.1	Brp3-F	GCGAGAGGTCACAGGAGAGC
	Brp3-R	CACTCGTTCTTCAGCCCTTG
MSTRG.2558.1	2558F	CCGTGTCAACCAAACCAGAAG
	2558R	CATTCAGACAAATCACACTCCACA
ROT75114.1	SC28-F	CGAGAAGGTGGGCGAACTGA
	SC28-R	CAGGTTGATGCCGATGGAGC
MSTRG.13279.1	13279F	GTCCACGGTAGTCAAGCAAGC
	13279R	TCTCTCTACCAGATGCTGTGAAAGT
MSTRG.23082.1	23082F	GAATGGACGGTGAAAACACTTATG
	23082R	TGCCTGGATGGGTAGGTGTG
ROT81705.1	Lys-F	TACTGGTGCGGAAGCGACTA
	Lys-R	GTAAGCCACCCAGGCAGAATA
ROT77417.1	E75-F	GACCAAGCATCACCCCGAGA
	E75-R	GCGGTGTGCTCAGTCATCTTGTA
ROT70303.1	Naa-F	CTACAACTCCTTCCGCCACAA
	Naa-R	ATGGCGAAGGTGGTGAAGC
ROT67869.1	GABA-F	CATTGGCTCCCCTCTGGTC
	GABA-R	CGAAGGTAGTCGGGGAACAA
ROT62527.1	62527F	CGAACGCAAAGGCAGCAC
	62527R	CGACCCTTGTTGCTTCCTCC
LOC113823636	Cactus-F	GCGGACGAAGACCTTGACT
	Cactus-R	GGCTGATGGTCATTCACTTGC
DQ206401.1	PEN2-F	GACGGAGAAGACAATGGAAACC
	PEN2-R	ATCTTTAGCGATGGATAGACGAA
DQ206403.1	Pen3-F	TACAACGGTTGCCCTGTCTCA
	Pen3-R	ACCGGAATATCCCTTTCCCAC
DQ206402.1	Pen4-F	GGTGCGATGTATGCTACGGAA
	Pen4-R	CATCGTCTTCTCCATCAACCA
AF430071.1	Cru1-F	GTAGGTGTTGGTGGTGGTTTC
	Cru1-R	CTCGCAGCAGTAGGCTTGAC
AY465833.1	Cru3-F	TCCACAATGGTCAGCGTCAAG
	Cru3-R	CTGTCCGACAAGCAGTTCCTC
EW713395	ALF1-F	TTACTTCAATGGCAGGATGTGG
	ALF1-R	GTCCTCCGTGATGAGATTACTCTG
EW713396	ALF2-F	GGCCATTGCGAACAAACTCAC
	ALF2-R	GTCCATCCTGGGCACCACAT
ABB22831	ALF3-F	CTCCGTGTTGACAAGCCTGG
	ALF3-R	GCAGCTCCGTCTCCTCGTTC

**Table 4 animals-13-01185-t004:** Non-specific immune indices in the serum of *L. vannamei* fed with increasing levels of CPC.

Parameter	Group			
	FM	CPC15	CPC30	CPC45
ACP (IU/g)	92.95 ± 0.92 ^a^	108.60 ± 3.17 ^b^	140.90 ± 2.40 ^c^	90.53 ± 1.72 ^a^
AKP (IU/g)	66.67 ± 1.32 ^a^	120.34 ± 4.52 ^b^	106.27 ± 3.78 ^b^	120.66 ± 2.77 ^b^
SOD (U/mL)	96.29 ± 0.54 ^a^	145.59 ± 3.28 ^b^	141.40 ± 4.23 ^b^	137.94 ± 3.24 ^b^
PO (U/L)	34.31 ± 0.64	34.30 ± 2.46	38.80 ± 2.12	36.14 ± 0.78
LZM (U/L)	2.92 ± 0.04 ^a^	2.92 ± 0.12 ^a^	4.90 ± 0.10 ^b^	4.26 ± 0.08 ^ab^
CAT (U/mL)	40.81 ± 1.21 ^a^	58.08 ± 1.32 ^b^	52.24 ± 0.38 ^b^	53.56 ± 0.47 ^b^
AST (mU/g)	44.75 ± 0.07 ^b^	27.35 ± 0.16 ^a^	29.16 ± 0.09 ^a^	44.41 ± 1.41 ^b^
ALT (mU/g)	74.78 ± 2.34 ^b^	72.75 ± 2.67 ^b^	58.59 ± 1.57 ^a^	88.91 ± 4.35 ^c^
MDA (nmol/mL)	10.70 ± 0.05 ^c^	6.36 ± 0.07 ^a^	8.04 ± 0.15 ^b^	7.93 ± 0.32 ^b^

Note: For each experiment, values with different letter suffix differ significantly (*p* < 0.05), and *n* = 3.

**Table 5 animals-13-01185-t005:** Statistics of transcriptome sequencing in *L. vannamei*.

Item	Raw Data (bp)	Clean Data (bp)	GC Percentage (bp)	Q20% (bp)	Q30% (bp)	Effective Reads (bp)	Total Mapped (bp)	Exon (bp)	Intron (bp)	Intergenic (bp)
**FM-1**	7,750,691,400	7,667,795,876	3,561,592,079 (45.95%)	7,500,430,776 (97.82%)	7,185,399,849 (93.71%)	51,526,760	41,699,138 (85.49%)	26,675,562 (63.97%)	3,204,123 (7.68%)	11,819,453 (28.34%)
**FM-2**	6,725,523,000	6,642,429,430	3,134,589,163 (46.61%)	6,481,225,567 (97.57%)	6,181,988,334 (93.07%)	44,694,126	37,060,603 (86.07%)	23,845,520 (64.34%)	2,734,914 (7.38%)	10,480,169 (28.28%)
**FM-3**	6,593,927,100	6,513,279,775	3,068,267,760 (46.53%)	6,358,061,273 (97.62%)	6,073,534,860 (93.25%)	43,825,148	35,434,954 (85.21%)	22,983,166 (64.86%)	2,730,666 (7.71%)	9,721,122 (27.43%)
**CPC15-1**	7,156,347,000	7,072,502,810	3,253,038,517 (45.46%)	6,906,298,032 (97.65%)	6,590,590,898 (93.19%)	47,571,384	38,572,370 (85.00%)	24,381,187 (63.21%)	2,984,675 (7.74%)	11,206,508 (29.05%)
**CPC15-2**	6,818,080,800	6,736,566,160	3,103,084,796 (45.51%)	6,573,194,948 (97.57%)	6,263,533,549 (92.98%)	45,334,742	36,034,745 (84.43%)	22,764,691 (63.17%)	2,787,220 (7.73%)	10,482,834 (29.09%)
**CPC15-3**	7,049,986,800	6,974,039,570	3,180,898,090 (45.12%)	6,808,099,463 (97.62%)	6,497,966,411 (93.17%)	46,870,694	37,195,349 (84.58%)	23,388,113 (62.88%)	2,996,288 (8.06%)	10,810,948 (29.07%)
**CPC30-1**	8,211,688,800	8,120,317,400	3,765,592,526 (45.86%)	7,933,918,636 (97.70%)	7,581,072,924 (93.36%)	54,588,588	44,193,568 (85.59%)	28,414,814 (64.30%)	3,391,737 (7.67%)	12,387,017 (28.03%)
**CPC30-2**	7,362,995,400	7,282,017,926	3,402,468,398 (46.21%)	7,106,052,798 (97.58%)	6,780,633,929 (93.11%)	48,933,996	40,421,630 (85.86%)	25,501,372 (63.09%)	3,078,600 (7.62%)	11,841,658 (29.30%)
**CPC30-3**	7,383,816,300	7,309,068,344	3,356,616,410 (45.46%)	7,142,935,453 (97.73%)	6,828,866,924 (93.43%)	49,102,968	39,634,331 (85.22%)	25,509,188 (64.36%)	3,152,083 (7.95%)	10,973,060 (27.69%)
**CPC45-1**	7,461,683,700	7,381,254,347	3,456,162,821 (46.32%)	7,212,569,018 (97.71%)	6,896,396,008 (93.43%)	49,604,658	40,140,871 (85.62%)	24,365,099 (64.84%)	2,849,807 (7.58%)	10,361,522 (27.57%)
**CPC45-2**	7,334,094,900	7,262,493,436	3,227,802,467 (44.01%)	7,111,478,739 (97.92%)	6,813,670,715 (93.82%)	48,787,044	37,576,428 (83.57%)	24,275,268 (63.78%)	3,000,806 (7.88%)	10,782,327 (28.33%)
**CPC45-3**	6,898,534,800	6,822,192,973	3,192,487,607 (46.28%)	6,650,991,010 (97.49%)	6,332,191,861 (92.82%)	45,841,774	38,058,401 (86.38%)	23,914,820 (65.83%)	2,895,969 (7.97%)	9,519,324 (26.20%)

**Table 6 animals-13-01185-t006:** Cactus genes and the related pathways.

Category or Gene ID	Gene Description	Species	Log2FC ^a^	Log2FC ^b^	Log2FC ^c^
**Toxoplasmosis**					
ROT61665.1	Cactus	*Penaeus vannamei*	−0.22	−9.16	−9.16
ROT61785.1	Hsc71	*Penaeus vannamei*	−0.61	−0.79	−0.87
ROT67561.1	Hsc71	*Penaeus vannamei*	−0.84	−0.89	−0.87
**Pathogenic *Escherichia coli* infection**					
ROT61665.1	Cactus	*Penaeus vannamei*	−0.22	−9.16	−9.16
ROT62386.1	Cycs	*Penaeus vannamei*	−0.84	−0.55	−0.54
ROT67928.1	Tuba1a	*Gecarcinus lateralis*	−1.74	−1.84	−2.48
ROT62891.1	Tbb-4	*Penaeus vannamei*	0.77	0.84	1.72
**Insulin resistance**					
ROT61665.1	Cactus	*Penaeus vannamei*	−0.22	−9.16	−9.16
ROT83272.1	Slc2a3	*Penaeus vannamei*	−1.50	−1.66	−1.23
ROT74164.1	Mlxipl	*Penaeus vannamei*	−0.69	−1.16	−1.01
**Prostate cancer**					
ROT61665.1	Cactus	*Penaeus vannamei*	−0.22	−9.16	−9.16
ROT63374.1	Hsp90b1	*Penaeus vannamei*	−0.92	−1.21	−1.22
**Toll and IMD pathway**					
ROT61665.1	Cactus	*Penaeus vannamei*	−0.22	−9.16	−9.16
ROT79438.1	Prss8	*Penaeus vannamei*	−0.92	−1.22	−1.21

Note: ^a^ denotes FM vs. CPC15 group, ^b^ denotes FM vs. CPC30 group and ^c^ denotes FM vs. CPC45 group.

## Data Availability

The data that support the findings of this study are available upon request from the corresponding author. The data are not publicly available due to privacy or ethical restrictions.

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
