# Peer review of "Transcriptome Analysis Reveals the Immunoregulation of Replacing Fishmeal with Cottonseed Protein Concentrates on Litopenaeus vannamei"

_animals, 2023, doi:10.3390/ani13071185_

Round 1

Reviewer 1 Report

This study investigated the effects of CPC substituting fishmeal on the immunological response and transcriptome of L. vannamei. The results can provide a theoretical basis for the replacement of FM with CPC in shrimp feed industry. I have the following comments.

L 17, instead of, revised as “substituting” or “replacing”.

L104-106, both units of % and g/kg were used here, it is confusing.

Table 1, Proximate composition, g/kg was used. Obviously, all the numbers were wrong. It is impossible. In dry matter, no moisture.

Table 2, the total was only 33-35 g/kg? It is impossible.

The growth performance is very important in feeding trial. Maybe the author had used the growth data in another paper. If it was true, please briefly describe the WG, survival and FCR before samples collection (Here, you could cite the paper, if the paper had been published). If it was not true, please add the growth data in RESULT section.

Line 137, How to collect blood and serum?

L 139, it is serum, or hemocytes for gene expression analysis? Please indicated it.

3.1. Non-specific immune indices in the serum. There are too many figures, and I suggest a table instead of figures.

L 413, CPC could have immune-boosting effects. Why? What components might be involved in such effects?

The English writing needs a careful check and revision.

 This paper needs major revision.

Author Response

Dear Reviewer,

Thank you for sending us the reviewers’ comments on our manuscript (animals-2234098) entitled “Transcriptome analysis reveals the immunoregulation of replacing fishmeal with cottonseed protein concentrates on Litopenaeus vannamei”. Those comments are valuable and most helpful for us to revise and improve our paper, as well as the important guidance to our research. We have now revised the manuscript according to the comments, and the revised parts in the manuscript were colored red. A detailed response to the comments is attached.

We would love to thank you for allowing us to resubmit a revised copy of the manuscript and we highly appreciate your time and consideration.

Kind regards,

Shuang Zhang

Reviewer 2 Report

In this study, the authors employed transcriptome analysis to study the immunoregulation of replacing fishmeal with cottonseed protein concentrates (CPC) on Litopenaeus vannamei. The results showed that the moderate amount of CPC substituted for fishmeal may improve the immunity of L. vannamei by suppressing the expression of Cactus genes, thereby increasing the expression of antimicrobial peptide genes. Overall, the work has innovation and the writing was very organized and clear. The following problems should be solved at first before the publication.

1.     “Fishmeal” and “fish meal” should be unified in the manuscript.

2.     Line 19: the definition of FM is not accurate, should not be defined fishmeal. change to 0% (control, FM) may be more accurate.

3.     Line 32: Please verify that “Toxoplasmosis” initials need to be capitalized. Please verify that “Escherichia coli” should be italicized.

4.     Lines 33 and 40: Reduce unnecessary abbreviations, e.g., IMD and AMP. other necessary abbreviations and their full names need to be given.

5.     The way references are cited in the main text is not uniform. There are references according to both numbers (lines 45-53, etc. ......) and names of people (55-56, 59, 337-338, 388-389, 417-418, etc. ...... not to be pointed out), check carefully to revise.

6.     Line 46: “And” does not need to be italicized.

7.     Line 65: The species in reference 15 is not “Epinephelinae”, please verify the correction.

8.     Line 92: “Vibrio vulnificus” should be italicized throughout.

9.     Lines 104-106: 0% g/kg, 15% g/kg, 30% g/kg, 45% g/kg. Units are confusing. Verify the revision.

10.  Lines 111-112: °C font format.

11.  Table 1:

Unit has been given in the table header and need not be reflected again in the table, except for some special units of composition. The dry matter basis has been defined in the table header and does not need to appear again in the table. DM and % should be separated by a symbol.

“Oil” in “Soybean Oil” does not need to be capitalized.

Is it necessary to capitalize “Mineral” in “Vitamin and Mineral premix”? It is recommended that “a” be superscripted.

The approximate ingredient unit is g/kg; please verify that the content of each ingredient is correct.

Is it necessary to capitalize Mineral in the table notes? A space should be added between the value and the unit.

12.  Please verify that the g/kg unit is correct. It is recommended to add a table note to list the full name of each amino acid.

13.  Line 128: “h” is a bit redundant.

14.  Lines 156 and 162: “TM” and “®” should be superscripted.

15.  Lines 157-161: It is not necessary to give the full name again for recurring abbreviations. It is recommended to italicize the gene names to distinguish between gene and enzyme activity.

16.  Line 213: Is it 9 antimicrobial peptide genes?

17.  Fig. 1 and Fig. 2: Is it necessary to capitalize “Serum” in the figure title.

18.  Lines 271-272: Spaces should be added between numbers/symbols and English. Similar situation in lines 388 and 433.

19.  Fig. 3: Latin should be italicized.

20.  Line 323: Same as question 3.

21.  Line 336: “P” should be italicized. The problem is not pointed out in the whole text, check and modify by yourself.

22.  Table 5: Two log2FCb?

23.  Line 346: Seven differential genes? Please verify the changes based on the results.

24.  Fig. 9: “QPCR” in the figure is suggested to be changed to “qPCR” to maintain the uniformity of the manuscript. Figure title “-ΔΔCt” should be superscripted。

25.  Line 412: Please verify that the CDCP is correct.

26.  Line 418: “X. Jiang et al., 2021”. Reference citation format error.

27.  Lines 413-414: CPC may have an immune enhancing effect on L. vannamei, as evidenced by the gene expression of several immune enzymes. The discussion of gene expression levels is a bit general. And the results presented at the gene level and enzyme activity level are not completely consistent, which should also be discussed slightly.

28.  Analysis of KEGG results showed that significantly enriched were IL-17 signaling pathway, Toll and IMD signaling pathways, NF-κB signaling pathway, PPAR signaling pathway, TNF signaling pathway and other immune-related pathways. Why the authors focused more on antimicrobial peptide related information.

Author Response

(The authors gave the same response as above.)

Reviewer 3 Report

Summary: The manuscript by Wang et al. explores evidence that cottonseed protein concentrates can alter immunological status in the economically valuable shrimp, Litopenaeus vannamei.

General comments: The title focuses on transcriptomic component of the work, but data from various other assays of immune function are reported. Background is adequately described. However, while this study appears to be correctly designed in terms of diet groups, number of animals per group, etc., the overall hypothesis underlying the study is not clear. Thus, the rationale for selecting transcriptomic analysis and the particular subset of other assays (immune markers, antimicrobial peptides) is not clear or justified. Data analysis is generally correct but presentation of the transcriptomic data results (particularly in Fig. 7 and associated text) is confusing both in terms of the figure and language. References are appropriately cited.

Specific comments:

Litopenaeus vannamei – more commonly known as Pacific whiteleg shrimp or Pacific white shrimp.

Page 1, Line 20-21 “At the end of the study, the activity and

expression of immune-related enzymes in the serum of the CPC-containing group showed almost the same significant trend as FM.” What does this sentence mean? 

Line 59: explain "depenalisation and moderate disaccharide"

Line 58 – 79 – this synthesis/review is difficult to understand.

Line 95-96: “transcriptome sequencing analysis was used to investigate the mechanism of CPC substitution for fishmeal on the immunological response of L. vannamei.” This statement of purpose is not clear, and should be stated as a scientific hypothesis. 

Lines 122-132: were water parameters measured continuously or periodically throughout the day (if so, how many times per day)? Was water re-circulated? Any mortalities? 

Line 133-140: Here and elsewhere: authors should define what they mean by the term “serum.” hemolymph, hemocytes and/or what other components? Elsewhere in the manuscript the text refers to “hemocytes” How was hemolymph extracted to ensure that loss from samples due to rapid clotting did not occur during the bleeding process?

Lines 133-140: Also, the text states “Eight L. vannamei were randomly collected from each group and mRNA extracted from the serum of each shrimp was mixed equally into one sample for immune gene expression and transcriptome sequencing.” If each diet regime is considered a “group” then there should be a total of 4 pooled samples. In contrast lines 189-195 refer to seven samples. Please clarify replication of samples, group, libraries, etc.

Lines 227-230: The text states: “The activities of ACP, PO and LZM in the serum of L. vannamei showed a trend of increasing and then decreasing with the increase of the CPC substitution ratio.” Suggest deleting this statement, since PO does not show the stated change; further ACP and LZM are discussed in the next sentence, which should also be edited to refer only to ACP and LZM.

Lines 230-232: The text states “However, AKP, SOD and CAT activities were substantially higher in the CPC group than in the FM group (P<0.05; Fig. 1B, C and F).” This statement should indicate that each of the three CPC groups had significantly higher values of these enzymes than the FM control. 

Lines 242-243: The text states (referring to Fig. 2) “the expression of PO, LYZ and ACP was substantially higher in the CPC30 group compared to the FM group.” This was not true for ACP.

Lines 245-246: The text states “Gene expressions of SOD1, AKP and AST were significantly higher in all CPC-containing groups than in the FM group (P<0.05).” Correct “all” to “most”

Line 287: Correct “EDGs” to “DEGs”

Line 302: Suggest changing “alternative” to CPC-fed or other more specific term. In fact, designation of diet groups should be consistent throughout the manuscript, to avoid confusing the reader.

Lines 296-324 and Figure 7: The text, diagrams and use of color in panels for Figure 7 are complicated and unfortunately confusing to the reader. Careful English language editing may solve some of these problems. Redesigning the figure, placing some of the data in supplementary tables, may help resolve some of these problems.

Lines 364-370: On what basis were these AMPs selected for quantification in Fig. 10? Were differentially expressed genes searched for association with any other immune-related functions?

Lines 376-385: The discussion here should be worded more precisely and carefully; this study documents some increases and other changes in expression and activity of some immune-related genes/enzymes. It does not support that these changes have a “positive” effect on immune defense, or that immune defense is “improved” with CPC-containing diet (Lines 448-453).

Author Response

(The authors gave the same response as above.)

Reviewer 4 Report

The study is well documented and structured, bringing novel and useful data for the aquaculture nutrition field and food safety overall.

The experimental designed is well-detailed and appropriate.

The Discussion and Conclusion sections are comprehensive and well-formulated.

Overall good English.

Only minor corrections, listed below:

Line 46. No need for italics at “and“ before “Penaeus“.

Line 48. Change the term “objects“

Line 67. Species name should be lower-cased: Trachinotus ovatus.

Line 71. Species name should be italized here.

Line 92. Species name should be italized.

Line 116. Table 1. The authors should explain why there are some difference in % of other compounds in the shrimp feed other than CPC and fishmeal (for example, soybean oil or cellulose).

Line 277. Use italics for the name of the species.

Line 287. EDGs???

Line 306. Use italics for the name of the species.

Line 323. Use italics for the name of the species.

Line 332. The resolution of Fig. 7 should be improved, it is not readable.

Line 342. Table 5 is repeated.

Line 426. Use italics for the name of the species.

I recommend publishing the manuscript with minor revisions.

Author Response

(The authors gave the same response as above.)

Round 2

Reviewer 1 Report

This paper has been revised, and it could be accepted except a small question as following.

Table 1, the formulating basis of diets is air dry basis, but not dry matter basis. For diet proximate composition, if you use air dry basis, you should supply the moisture level in diet. Dry matter basis means no moisture.

Author Response

Dear Reviewer,

Thank you for sending us the reviewers’ comments on our manuscript (animals-2234098) entitled “Transcriptome analysis reveals the immunoregulation of replacing fishmeal with cottonseed protein concentrates on Litopenaeus vannamei”. Those comments are valuable and most helpful for us to revise and improve our paper, as well as the important guidance to our research.

We would love to thank you for allowing us to resubmit a revised copy of the manuscript and we highly appreciate your time and consideration.

Kind regards,

Shuang Zhang

Reviewer 2 Report

I think the manuscript can be published in its current state.

Author Response

(The authors gave the same response as above.)
